# Underground Nests and Foraging Activity of Invasive Conehead Termites (*Nasutitermes corniger*; Blattodea: Termitidae)

**DOI:** 10.3390/insects16121262

**Published:** 2025-12-12

**Authors:** Barbara L. Thorne, Katherine E. Tenn, Sue Alspach, Monica N. Roden, Marah S. Clark

**Affiliations:** 1Department of Entomology, University of Maryland, College Park, MD 20742, USA; 2Florida Department of Agriculture and Consumer Services, 3125 Conner Boulevard, Tallahassee, FL 32399, USA; katherine.tenn@fdacs.gov (K.E.T.); salspach58@hotmail.com (S.A.); monica.roden@fdacs.gov (M.N.R.);

**Keywords:** Nasutitermitinae, invasive termite, below-ground nest, invasive eradication

## Abstract

Conehead termites (species *Nasutitermes corniger*) are ecologically important decomposers in their broad native range of Central and South America as well as most Caribbean islands. They can also be pests of agriculture, structures, and invaded natural areas. Breeding populations derived from introduction of a single *N. corniger* colony were first discovered in south Florida, U.S.A. in 2001. Florida’s Department of Agriculture and Consumer Services developed an effective, science-based approach to suppressing conehead termite infestations with the goal of eradication. A key strategy has been to locate and destroy the conspicuous above-ground nests typically built by *N. corniger*. Nests are the heart of a colony, housing reproductives, white immatures (instars 1–3), and a large portion of the population. This paper describes the surprising discovery of underground invasive conehead termite nests, documenting a previously unknown habitat for this adaptable, economically important species. Characterizing partially and completely underground *N. corniger* nests will enable inspectors, pest managers, and researchers to find, remove, and manage conehead termites.

## 1. Introduction

The termite genus *Nasutitermes* is broadly characterized by above-ground nesting and foraging. *N. corniger* (common name ‘conehead termite’) is the type species of this genus and the most widely distributed, most abundant termite species in the Neotropics. Across its range from Mexico through Central America, at least 11 countries in South America, the Caribbean islands, as an established invasive species in New Guinea, and as a recent invader in Florida, USA, *N. corniger* is known as a primarily arboreal species, also building mounds and foraging tunnels on the ground surface (reviewed in [1,2]).

Older *N. corniger* colonies build dark brown carton nests, usually roundish or ellipsoidal with a bumpy surface, reaching approximately 1 m in height [3] (Figure 1). Nests may be on, in, or by a tree (including high in the canopy [4]), shrub, grass clump, stump or log, on the ground, in litter, or on or in a structure [1,2,4,5,6,7,8,9]. Although generally built on or above-ground, conehead nests and foraging galleries can extend shallowly below the soil surface with the majority of the structures above-ground level [1,2,10]. In addition to typical nest and foraging tunnel sites on or above-ground, largely or completely underground *N. corniger* nests and tunnels were discovered in two locations in Broward County, Florida. This paper documents invasive *N. corniger* activities below-ground in south Florida.

Describing the location of these structures and activities raises potential confusion due to termite terminology. It is important to clarify that, because the phrase “*subterranean termite*” refers to three genera of economically important termite taxa (*Reticulitermes*, *Coptotermes*, and *Heterotermes*) that are “ground-dwelling” without visible nests or mounds [11] (p. 141), we intentionally use alternative wording to specify site locations in this report. In no way do we propose that *N. corniger* be categorized as a “subterranean termite.”

To prevent misinterpretation of *subterranean* as an accurate site location rather than as a category of life history, we never use the phrase “subterranean termite” in reference to *N. corniger*. Furthermore, although we consider *subterranean*, *underground*, *below-ground*, and *subsurface* as synonyms describing when all or some of a nest or foraging activity is beneath the soil surface, in this paper we use alternatives to ‘subterranean’ to reduce confusion.

The following additional definitions pertain to this paper. “*Completely*” or “*entirely*” underground means that all of the nest carton is below the soil surface, sometimes with a shallow ‘crown’ extending less than 3 cm above ground level. This includes nests with their top carton flush with or below the soil surface. Completely underground nests may also be built with their top under and abutting a partially, or entirely, buried substrate such as a rock, root or root ball, or stump. A nest mound with carton built more than 3 cm above-ground and also extending galleries into soil, perhaps with the majority of its structure subsurface, is characterized as *partially* underground.

Both locations harboring substantial subsurface invasive conehead termite activity in Florida were on or near canal banks that sloped sharply up from water level, rising to flatter shore embankments approximately 0.5–2 m above low tide depending on location and tide cycle. The canal bank up to, and extending across, the flatter topography along the embankment housed typical above-ground as well as partially and completely subsurface *N. corniger* nests and foraging tunnels. Underground nests or subsurface portions of nests in each site sometimes had sections of carton adjacent to, or built partially into, an air space created by gaps between tree roots, piles of rocks, broken cement slabs, or combinations of those features below-ground level. In such cases a nest is still largely surrounded by soil but is not completely enclosed within soil due to abutment with an airspace. We term these circumstances as *underground open void* nest positions.

Invasive *N. corniger* were first discovered in the United States, in Dania Beach, Broward County, Florida, in 2001 [12]. The Florida Department of Agriculture and Consumer Services (FDACS) invasive conehead termite eradication program, intensified since 2012, has worked in diverse infested areas and habitats in Dania Beach, Davie, Pompano Beach, and Ft. Lauderdale (all in Broward County, Florida). Locations vary across natural, residential, and commercial properties; some wooded, others with managed landscapes; mangroves; and some along other canals or roadsides [1,2,13,14,15,16]. Across that range of *N. corniger* infested sites we did not encounter substantially or fully underground nests until thoroughly exploring two sites bordering the Dania Cut-Off Canal, one in Dania Beach, the other in Ft. Lauderdale.

This paper describes and illustrates below-ground structures built and occupied by invasive *N. corniger* in south Florida. We discuss site characteristics and patterns that may have contributed to the conehead termites exploiting subsurface habitats, along with how to inspect for and treat underground invasive activity. This report expands previously known nesting and foraging gallery location options, further broadening understanding of the ecological plasticity and adaptability of *N. corniger*.

## 2. Materials and Methods

### 2.1. Sites

Underground *N. corniger* activity in south Florida occurred primarily in two locations bordering the estuarine Dania Cut-Off Canal, east of highway Interstate 95, Broward County, FL. One site, “Airport,” is along the Canal parallel to the southwest portion of South Perimeter Road, Fort Lauderdale-Hollywood International Airport. The second is along Old Griffin Road, Dania Beach. Both locations feature a narrow strip of vegetation 5–10 m wide between the canal banks and transit pavement [the perimeter road near airport runways (Airport; GPS 26.065737, −80.161028) or a busy vehicle traffic road (Old Griffin Road; GPS 26.060836, −80.157638)]. The two sites are about 0.35 km apart at their closest points. The *N. corniger* infestation spans approximately 400 m along the canal at the Airport and extends 325 m along the canal aside Old Griffin Road.

A single nest with its top flush with the ground surface under a large concrete slab, with a 15 cm diameter nest in soil under the slab, was found in 2024 at a third site, in Ft. Lauderdale (GPS 26.067410, −80.164648), ~30 m east of Interstate 95, and ~0.15 km west of the subsurface *N. corniger* discoveries at the Airport. This third site, “Ft. L I-95,” was ~60 m south of the Dania Cut-Off Canal, not on the canal’s banks as at the Airport and Old Griffin Road sites. The canal bank sites are about 4.5 km inland (west) from the Intracoastal Waterway and the Atlantic Ocean, directly across the Dania Cut-Off Canal from the Airport site. The “Ft. L. I-95” nest was located within a 12 hectare overgrown site that had no on- or above-ground *N. corniger* colonies, but our eradication program identified it as high risk for colonization given its proximity to known swarming over two decades.

### 2.2. Canal Bank Structure and Vegetation

The Ft. Lauderdale Airport and Old Griffin Road sites each have sandy soil banks along the Dania Cut-Off Canal, with tree roots and lower limbs exposed within intertidal level and above to the bank surface. Depending on tide height, the top of the bank is 0.5–2 m above water level at the Airport and along Old Griffin Road.

Thick, twisted roots and branches create open voids and cavities in the soil anchoring trees. Undercuts and contours occur—and change over time—in both canal bank’s soil due to erosion and silt from tide, boat, and storm surge currents. The Airport site has large trees including sea grape (*Coccoloba uvifera*), tropical almond (*Terminalia catappa*), Brazilian peppertree (*Schinus terebinthifolius*), red mangrove (*Rhizophora mangle*), sabal palm (*Sabal palmetto*) and coconut palm (*Cocos nucifera*). Some of the undergrowth plants are previously dense thickets of coinvine (*Dalbergia ecastaphyllum*), various tall grasses, and weedy forbs. To aid in the invasive termite surveillance and treatment protocols, regular maintenance on the densest areas of undergrowth have occurred in the past several years leaving zones of pruned, decomposing vegetation and leaf litter on much of the Airport bank surface.

Sections of the Airport bank contain large cement blocks and rocks, many broken into irregular chunks buried and embedded haphazardly in the soil, likely deposited as sturdy debris to support the bank structure. The blocks and rocks in and on the soil create surfaces, as well as gaps and channels, in the erratic piles where they were dumped. Spaces between adjacent pieces of block and stone may be open air or filled, completely or partially, with sandy soil, roots, and/or litter debris.

The canal bank along Old Griffin Road has a similar vegetative and structural composition. It also hosts many large Australian Pine (*Casuarina* sp.) and green buttonwood (*Conocarpus erectus*) trees among primarily red mangroves. The bank contains extensive networks of intertwined roots and areas of thick humus leaf litter, especially under the large pine trees. Some areas have substantial scattered limestone boulders and trash debris embedded in the soil.

*N. corniger* use these structural features of the canal banks to build opportunistically within soil and wood substrates, often constructing nest and foraging tunnels in crevice and cavity spaces as well as under or abutting roots, cement blocks, and/or rocks.

### 2.3. Research Approach Within Invasive Termite Eradication Areas

The Florida sites highlighted in this paper are among several locations infested by invasive *N. corniger*, thus are a part of FDACS’ on-going eradication program [1,2,13,14,15,16]. The elimination goals and protocols central to FDACS’ initiatives inform policy that no field experiments designed with replications and—especially—undisturbed control treatments that might grow and disperse, are permitted in these areas (see rationale in [16]). Experiments should be performed in areas where the species is permanently established rather than where an invasive population is possible to eradicate. This paper thus summarizes observations and data regarding underground *N. corniger* activities which were removed, destroyed, and treated with insecticides soon after discovery.

## 3. Results

### 3.1. Number of Entirely- or Partially Below-Ground Conehead Termite Structures Found

FDACS’ conehead termite eradication program’s first observations of subsurface *N. corniger* nests occurred in spring 2024. These discoveries prompted survey protocol changes in 2025 to be aware of, locate, and record data pertaining to underground activities (see Section 3.8). Data reported here are thus primarily from 2025, including 2024 descriptions in cases where we had explicit records in that season’s field notes.

Thirty-seven substantially or completely underground nests were discovered, removed, and treated at the Airport and Griffin Road sites in 2025. Each carton structure housed eggs and/or instars 1–3, thus being a true nest. A total of 20 of these were completely underground (nest apex < 3 cm above ground surface); 13 were partially underground [Table 1]. Four nests were entirely or partially below ground but missing specific data thus uncategorized. Subsurface foraging tunnels occurred associated with underground nests at each site. See Appendix A for raw data collected on each underground *Nasutitermes corniger* nest discovered and extracted in Broward County, Florida, USA in 2025.

### 3.2. Colony Population Within Underground Nests

All data presented here, and most subsurface carton structures discovered in south Florida, were true nests, containing eggs, white immatures (instars 1–3), soldiers, and workers; often alate-derived Queens and Kings, and occasionally nymphs of developing alates. Nymphs with wingpads (late stage alate nymphs) were found rarely in 2025, possibly due to the intensity of the 2024 eradication treatments. Most nests discovered in 2025 were young colonies not found the previous year, or small nests rebuilt by survivors of the previous year’s treatments. Both of those circumstances are characteristic of colonies not yet mature enough to produce alates. Several of the smaller carton formations found below ground were ‘foraging centers,’ rather than true nests, meaning that they housed only workers and soldiers with no evidence of a royal cell or any other structural differentiation. Foraging centers are omitted from data reported in this paper.

### 3.3. Size and Depth of Entirely- or Partially Underground Nests

Conehead termite nests from approximately 5–40 cm diameter and 2.5–40 cm in height were found under the soil surface. Those metrics represent the full range of sizes found in above-ground nests across the invasive populations in south Florida (Dania Beach, Pompano Beach, Ft. Lauderdale (Table 1, Figure 2 and Figure 3).

### 3.4. Construction: Underground Nest Carton Density, Texture, Composition, Proximity to Wood

The cartons of most underground *N. corniger* nests closely resembled classic carton characteristic of above-ground structures, including similar gallery size, dark brown color, carton composition containing abundant partially digested wood materials (though sometimes with more sand and dirt inclusions), and with interior structure reinforced and hard (Figure 2a). Some smaller nests, particularly those categorized as partially underground, contained more dirt or sand than in a typical above-ground nest, rendering the carton texture more fragile. In some cases, the sandy carton visible above-ground disintegrated when touched lightly (Figure 2f). Royal cells with Queen chamber(s) were found in several nests.

The surface of underground nests is bumpy like above-ground exterior carton. Below-ground nests or portions of nests may have multiple, connected carton lobes or extensions built in soil or air voids amidst roots, rocks, and cement blocks. Such lobes may be entirely underground, or at points adjacent to air pockets within rock piles, for example, or opening onto the canal bank.

Young on- or above-ground dwellings of *N. corniger* colonies typically begin with the royal pair (or multiple alate-derived Queens and/or Kings) hidden within wood [5,8,17,18,19,20]. As the colony grows, its first stage of carton nest construction yields a structure about the size of a fist, usually adjacent to or near the original royal cell in host wood. Healthy colonies continue to expand their nest such that it can become the size of a watermelon in as quickly as a few months, producing alates that year [1,16,20]. Note that Constantino [21] (p. 614) says “Nests of nasute termites … are usually not in immediate contact with wood.” That has not been our experience with *N. corniger* in Florida nor in its native range; nests are generally built on or surrounding wood substrate, although older colonies may have fully consumed that original perch [2,5,6,18,20].

We do not know the ontogeny of subsurface *N. corniger* nest development, but many of the entirely below-ground nests excavated in this study were anchored to an underground root, stump, or log, similar to above-ground construction. Once the termites consume the initial wood and a royal cell is constructed within the nest, no wood inclusion remains and a nest may appear to be ‘free floating’ in soil or anchored on at least one side to a rock or cement slab substrate.

### 3.5. Underground Foraging Activity

*N. corniger* subsurface foraging tunnels are constructed with size (usually ≥1.3 cm width) and texture similar to above-ground *N. corniger* tunnels. They are built of carton, with more sand and soil inclusions than typical of arboreal carton tunnels, but still sufficiently durable to be felt with a gloved hand carefully searching underground for termite structures. Underground foraging tunnels are built along roots, rocks, and concrete substrates, as well as unanchored, surrounded by soil. As an example, we tracked an active *N. corniger* foraging tunnel more than 2.5 m in length, 23–31 cm below the soil surface, constructed on a cement slab at the Airport site.

### 3.6. How to Find Below-Ground N. corniger Structures

Underground activity was discovered by experienced invasive conehead termite field team members who noticed lots of active, above-ground foraging tunnels in these areas, but no visible nests. That circumstance is reasonably common even within *N. corniger’s* “native” range. The termites may nest hidden within a tree trunk, branch, stump, or root. In the Airport and Old Griffin Road habitats, however, and within a wide radius of the Ft. L I-95 site underground nest, suspicions of novel nesting locations were raised following careful inspections.

In one striking example, active tunnels at the base of a tree contained nymphs with wingpads (late stage alate nymphs) but with no signs of a nest in the vicinity. Despite abundant active foraging tunnels, meticulous examination of above-ground wood locations unveiled no signs of termite occupation within that wood. Proceeding with the logical approach of “follow the termite activity,” the field team found hot spots of *N. corniger* termite foraging traffic at or near soil level. Busy hubs of workers and soldiers were revealed by gently clearing away ground vegetation, including leaf litter, to expose the surface as much as possible. Excavating and exploring in the direction of the most active areas, including holes into the soil (Figure 4), ultimately exposed extensive, lively underground nest and foraging tunnel structures (this ‘Under Sea Grape Tree’ nest (Figure 2a) is highlighted in Section 3.7).

Field experts advise that in a quest for nest locations, including those partially or completely below the soil surface, observe active, above-ground *N. corniger* sites from all directions and angles. Shadows and objects in the landscape can obscure foraging tunnels and nests viewed from particular vantage points. All possible sites above, at, or beneath soil level must be considered. Lift substrates such as rocks, stumps, or debris because many underground nests were discovered under large objects (Table 1, Figure 5).

Carefully differentiating subtle soil textures and density with your fingers, even through gloves, can be immensely helpful. When exploring loose or sandy soils, and if safe from buried glass, sharp metal, and other perilous dangers including animals, excavate gently with shovels, then your hands—covered with thin gloves if necessary—to feel at angles that trowels cannot reach. Look and feel for changes in texture that differentiate from surrounding soil to locate denser, rough nest carton and the harder shells of foraging tunnels. Normal soil often moves and can be easily shifted. *N. corniger* carton is discernably more rigid; it is ‘glued’ together with fecal material including lignin. The differences in texture and density composition of soil vs. carton can be valuable and practical in locating termite nests. Tools such as a screwdriver, rebar rod, or flexible probe can be helpful to distinguish differences in density and sound to locate additional underground carton.

### 3.7. Examples of Underground N. corniger Nests

The following descriptions illustrate some of the breadth of positions and circumstances of underground *N. corniger* nests discovered in 2024 and 2025.

#### 3.7.1. Under Sea Grape Tree

The location of this nest (also mentioned in Section 3.6) was narrowed after noticing nymphs of developing alates in foraging tunnels on a sea grape tree, without an associated nest detected. The canopy of this large tree generated 5–10 cm of thick leaf litter on the ground, limiting standard survey visibility of the soil surface. Three people worked multiple days to clear litter and ultimately expose the nest site with its top at ground level. The nest top was not detected visually. It was discovered by the ‘follow the termites” strategy of observing areas of relatively high forager traffic on the soil surface, including a careful tactile survey of the ground. The precise location of the nest was revealed by using hands or gentle probing with tools to discern soil density and texture differences distinguishing the harder nest carton. The 40 cm diameter nest, embedded with twigs, roots and leaf debris, extended 15 cm below the soil surface (Figure 3a and Figure 4). That nest housed at least nine alate-derived Queens, along with mature alates, late-stage alate nymphs, eggs, white immatures instars 1–3, workers, and soldiers.

#### 3.7.2. Ft. L I-95

The “Ft. L. I-95” site involved a single, isolated nest discovered by noticing active foraging tunnels on small trees. No above-ground nests were visible; tunnels tracked under a large concrete slab lying on the ground surface. After large machinery lifted the concrete, the top of a carton nest was exposed (Figure 5f). The nest crown abutted the concrete at ground-surface level, with the carton structure extending approximately 10–15 cm under the soil. A monogamous pair of an alate-derived Queen and King were found in the nest along with workers, soldiers, eggs, and white immatures (instars 1–3). Of the several hundred invasive *N. corniger* nests discovered in Florida since the renewed eradication effort began in 2012, this is one of very few colonies found with only a single small Queen and King, consistent with this secluded nest being founded by dispersing alates from established infestations.

#### 3.7.3. Deep Under a Boulder

Among our discoveries, the *N. corniger* nest found the farthest below the soil surface abutted a large limestone boulder at Old Griffin Road. There were small areas of carton structure and some termite activity along the junction between the soil surface and the side of the embedded boulder. After moving the boulder, a large nest approximately 20 cm wide and 10 cm tall was discovered with its deepest point 45 cm below-ground where the boulder had rested within the soil (Figure 5a,b). The nest contained workers, soldiers, eggs, and white immature (instars 1–3) termites. No reproductives were retrieved, but the existence of eggs suggests that Queen(s) and King(s) had been present in the colony but escaped during nest extraction.

#### 3.7.4. Canal Bank, Abundant Queens

A partially below-ground nest was discovered embedded into the side of the canal bank at the Airport site (Figure 2a,b). This nest was partly exposed above soil with 15 cm of round carton visible emerging from a tree root ball. The volume of the extracted underground nest structure, including from voids deep within the canal bank, totaled greater than a basketball in size. Over 100 alate-derived Queens were in the removed carton, the highest number of Queens found in any nest treated in 2025 (see [1,2,8,19] for information on polygyny and polyandry in *N. corniger*).

### 3.8. Best Practices for Removing, Destroying, and Treating Underground Invasive N. corniger

FDACS’ invasive conehead termite eradication program field team has proven success implementing the same integrated pest management (IPM) principles in treating below-ground as well as above-ground activity (detailed in [1]). The approach of diligent surveys to locate nest and foraging sites, extraction followed by crushing nest materials, and immediate treatment of contiguous hard surfaces bordering the nest (wood, rock, cement) as well as crushed nest debris with appropriate, precisely targeted insecticides works effectively. In most cases, the sandy soil where the underground nests were removed was not treated with insecticide because it was soft and would quickly be dislodged by wind or rain. Follow-up monitoring is required, as with treatments in above-ground habitats, in case some reproductives escape interventions and rebuild their colony.

Below-ground discoveries, nest extractions, and treatments can be more time-consuming because they require comprehensive exploration and destruction of potentially multi-lobed nests built in inconvenient spaces under and between heavy rocks and cement blocks as well as immoveable tree roots and stumps. Diligence to approach the site from multiple directions enables the most thorough (complete) nest removal.

## 4. Discussion

*Nasutitermes corniger* thrive in diverse habitats including tropical forests, secondary growth, savannahs, mangroves, as well as among human structures and landscapes. This ecologically agile species is often the most prevalent, ‘dominant’ termite species in a community assemblage [4,5,6,22,23]. Because of their broad diet, ability to thrive in a wide array of habitats, and large population sizes, *N. corniger* are keystone decomposers for nutrient cycling via cellulose digestion. They are thus influential if not essential for self-sustaining natural ecosystems within their native range (fitting criteria for Paine’s definition of a keystone species, [2,24,25,26].” This report documents that in addition to their already known broad habitat use and important ecological and economic influences, *N. corniger’s* territory and impacts can extend underground as well.

Although generally built on or above ground, there are rare, brief reports documenting active *N. corniger* nests and foraging galleries extending shallowly (20 cm or less) below the soil surface, with the greater part of the structures above ground level [1,10]. Emerson [17] (p. 262) reports subsurface foraging tunnels of *N. corniger*. He states, “The covered tunnels may lead down into the ground where the excavations are lined with carton … Covered tunnels on the walls of Chilibrillo Cave, Panama (in what is now Parque Nacional Chagres), made by *N. corniger*, were estimated to be about twenty feet below the surface of the ground.” The wording does not explicitly detail whether those foraging tunnels were built completely or partially within soil, or if they are entirely or partly on rock or other substrate surfaces in air voids extending 20 ft (6 m) below ground level. Dudley [4] and Snyder and Zetek [10] also mention “underground” *N. corniger* foraging tunnels without elaboration. Zorzenon and Campos 2015 [27] include observations of “subterranean” activity of *N. corniger* in São Paulo, Brazil, but contains no mention of entire nests underground.

In addition to the sites highlighted in this paper, partially underground nests were observed infrequently at other sites in Broward County, Florida, including ≤10 cm below the soil surface under epigeal nests (N = 3), and slightly deeper (≤20 cm) beneath nests built amidst fakahatchee grass roots (N = 2), in Dania Beach and Pompano Beach, FL. Nests built into the tops or exposed sides of palm tree root balls, extending indeterminately deep into the dense root matrix, were found in other Florida locations and at the Airport and Old Griffin Road sites.

Because of the thorough inspections and monitoring required for the eradication effort, we are confident that completely or extensively below-ground nests were not present in other conehead-infested areas of Broward County, Florida. We did not treat subsurface locations outside of the sites described above, yet we achieved long-term suppression of populations at other sites (no live termites found for ≥5 years), which is another line of evidence that underground nests did not occur in those locations. Substantial below-ground activities of *N. corniger* may occur within the species’ natural range but remain to be discovered.

All *N. corniger* in Florida are invasive. Genetic analyses confirm they are descendants of a single colony introduced before first discovery in 2001 ([16]; Thorne and Vargo unpublished data). Colonies and populations are thus highly inbred. The history and novel circumstances of *N. corniger* in Broward County, Florida, may contribute to their exploiting habitats, including underground resources, in different ways.

Observed habitat patterns and commonalities of the Airport and Old Griffin Road sites, especially features that contrast with other infested areas without subsurface activity, may offer insights into why those two locations harbored underground *N. corniger* nests and tunnels. The Airport and Old Griffin Road locations shared the characteristics of bordering the Dania Cut-Off Canal, with rising banks and strata of sandy soils embedded with infrastructure and air voids created by exposed thick roots and chunks of hard materials (concrete/cement blocks and rocks at the Airport; rocks and debris in the soil along Griffin Road). Large blocks and rocks provide substrates and shield/create/protect voids; they also influence microsite temperature and moisture regimes.

Throughout Florida’s *N. corniger* eradication program, no other upland properties received as many consecutive years of treatments as the two sites featured in this report. Treatments in early 2025 were the fifth consecutive year in which all discovered nests were destroyed, and above-ground nest locations (the only nests identified in 2021, 2022, and 2023 at these sites) were treated with insecticide, at the Airport and Old Griffin Road. Previous treatments of infested properties in other locations ranged from 1 to 4 years in duration, sometimes not annually consecutive. Perhaps the continuous pressure of removing and chemically treating above-ground *N. corniger* activity at the Airport and Old Griffin Road favored persistence of colonies that exploited underground habitats.

The Airport and Old Griffin Road locations are impacted by loud noises, and their associated vibrations, caused by airplanes, vehicles, and boats. Other Broward County locations infested by invasive *N. corniger* were also proximate to transit and construction sounds yet harbored no underground nests that we found. Furthermore, the sites featured in this report also hosted above-ground nests, so we do not think that loud noises were decisive in promoting subsurface *N. corniger* activities.

Identifying circumstances that are hospitable to *N. corniger* activities will expand understanding and prediction of the species’ abilities to occupy underground habitats. That knowledge will facilitate discovery and control of invasive populations, and broader awareness of this adaptable termite’s potential ecological options and impacts across its range.

## 5. Conclusions

Beneath-ground-surface nesting and foraging activities are not rare for *N. corniger* in the sites highlighted in this paper. Subsurface habitats are yet another option for this opportunistic, flexible, and remarkably adaptable termite to exploit in their natural range or as an invasive species. *N. corniger* is the only known termite species that can nest and thrive at any level from high up in a tree canopy to the depth of at least 55.9 cm (22 in) underground. Descriptions and characterizations in this paper detail general patterns and observations regarding subsurface nesting and foraging. They underscore the need to be alert to exceptions and novelties when searching for *N. corniger* while conducting basic research or applied control initiatives. All inspection and treatment recommendations described above reflect best practices and lessons learned from the Florida Department of Agriculture and Consumer Services’ invasive conehead termite eradication program.

## Figures and Tables

**Figure 1 insects-16-01262-f001:**
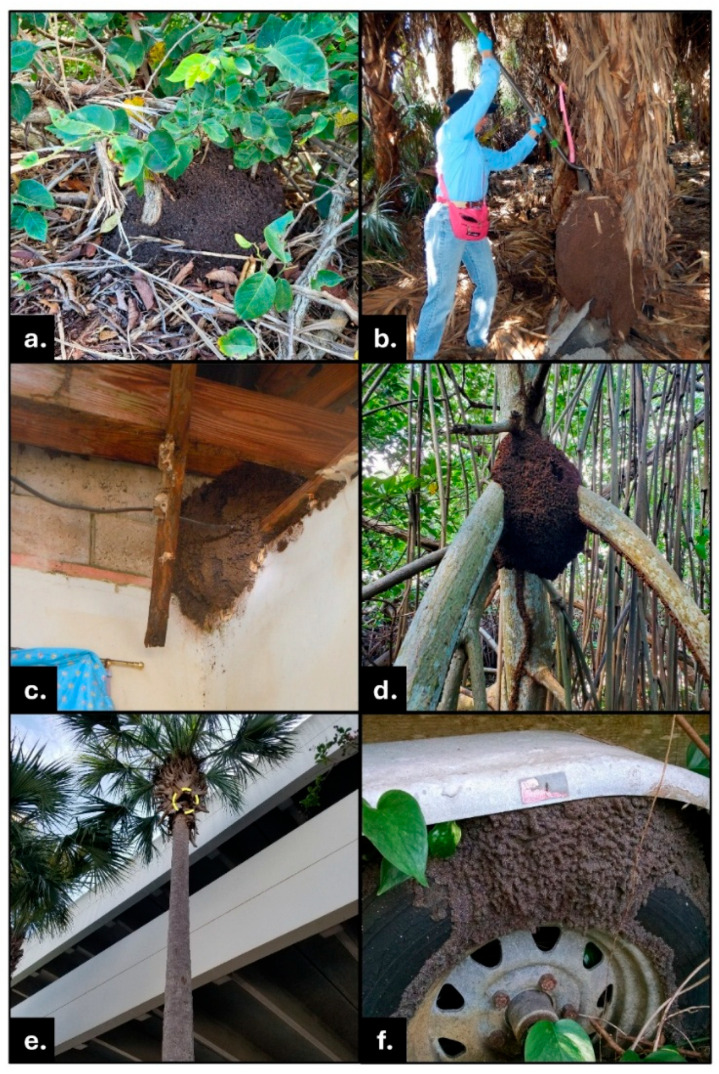
Examples of typical above- and on-ground *N. corniger* nests built in various circumstances in Broward County, Florida. (**a**) Above-ground surface (epigeal) nest amidst coinvine (*Dalbergia ecastaphyllum*); (**b**) Large nest on palm tree (*Sabal palmetto*) trunk, removal by FDACS eradication team in progress; (**c**) Nest in roof rafters of house; (**d**) Nest, with foraging tunnels visible, surrounding trunk and prop roots of red mangrove tree (*Rhizophora mangle*); (**e**) Nest (in yellow dashed circle) 6 m (20 ft) high in palm tree (*Sabal palmetto*) crown; (**f**) Nest constructed in wheel well and on tire of a boat trailer.

**Figure 2 insects-16-01262-f002:**
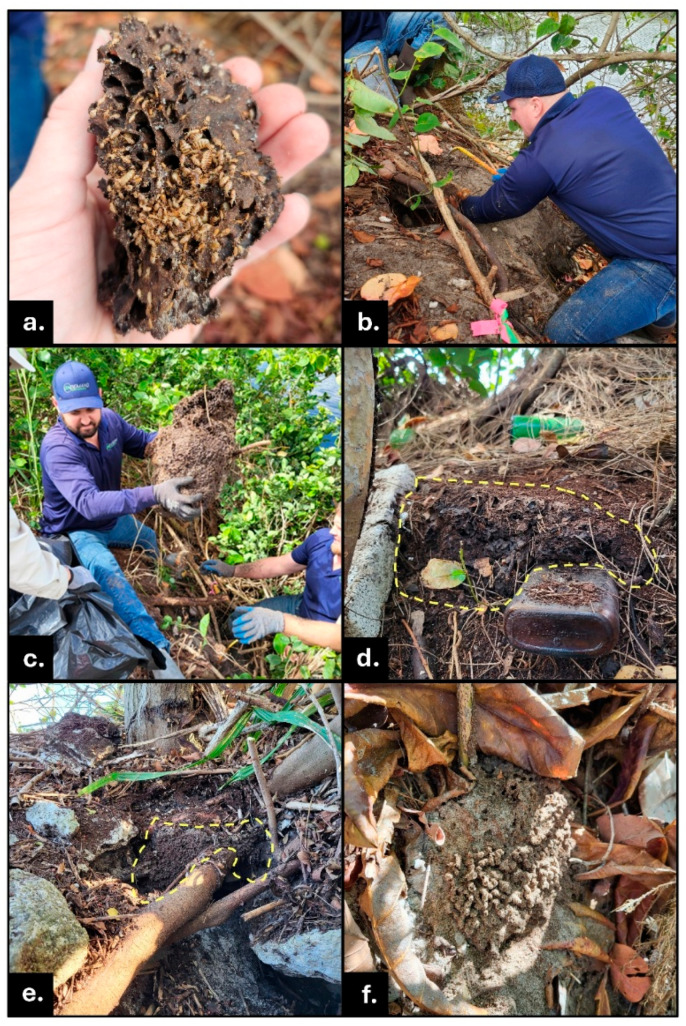
*N. corniger* nests found with at least some portion of carton visible on the ground surface, found on slopes of the canal banks described in this report (Broward County, FL). (**a**) Nest carton removed from the completely underground nest shown in the hole in photo (**b**). Note the typical carton composition and gallery architecture of this nest piece despite being underground. Photo (**a**) also shows multiple Queens (this nest housed more than 100 alate-derived Queens), workers, soldiers, and white immature (instars 1–3) termites. (**b**) Removing underground nest from void in canal bank in soil between tree roots. (**c**) Large underground nest extracted from heavily vegetated canal bank; the nest carton surrounds tree roots (now cut). (**d**) Exposed carton of nest (highlighted with yellow dashed line) embedded in canal bank, glass bottle debris in lower right of photo. (**e**) Nest carton (highlighted with yellow dashed line) in canal bank partially embedded in soil and in voids between tree roots and concrete riprap. (**f**) Top surface of soil nest in sand. Sand grains visible in carton material.

**Figure 3 insects-16-01262-f003:**
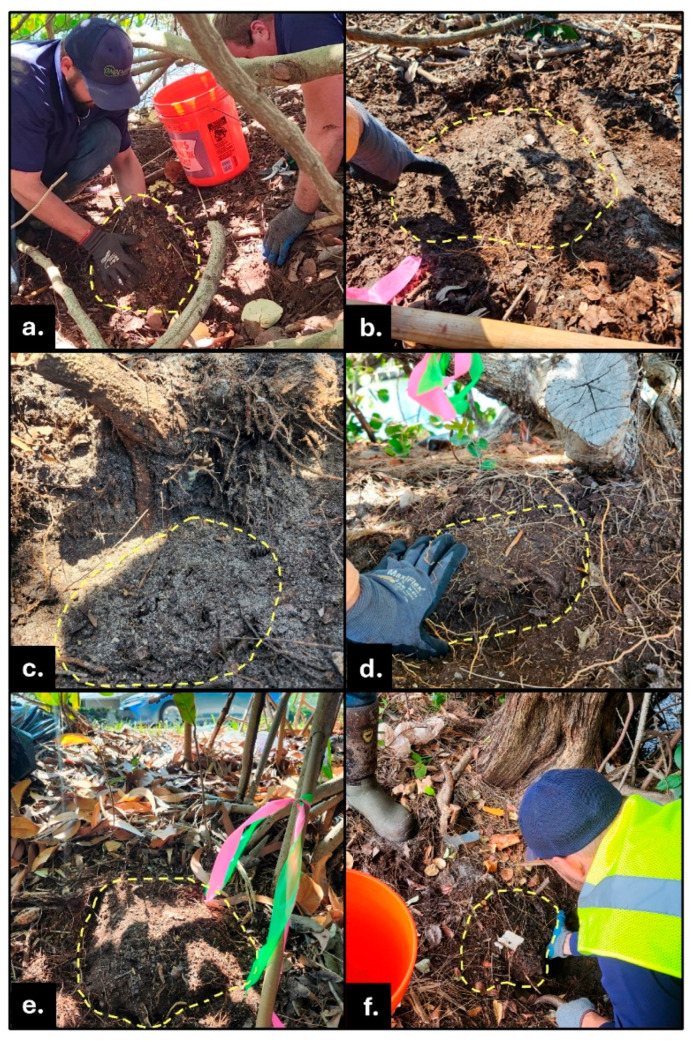
Completely underground *N. corniger* nests with their top surfaces visible after loose litter was swept away. These nests were constructed under horizontal ground, not on the canal bank slopes. Nest carton is outlined with a yellow dashed line. (**a**) Underground nest beneath canopy of a sea grape (*Coccoloba uvifera*) tree; removal in progress before treatment. (**b**) Exposed dome-shaped top surface, “crown,” of underground nest. (**c**) Underground nest surface revealed under tree roots in sandy soil. (**d**) Underground nest partially exposed in dense root ball. Top ground surface is shown at center of photo; soil in the foreground was removed to expose the nest. (**e**) Upper surface of nest built underground beneath leaf litter amid red mangrove (*Rhizophora mangle*) roots. (**f**) Underground nest’s top between roots of green buttonwood (*Conocarpus erectus*); nest removal in progress.

**Figure 4 insects-16-01262-f004:**
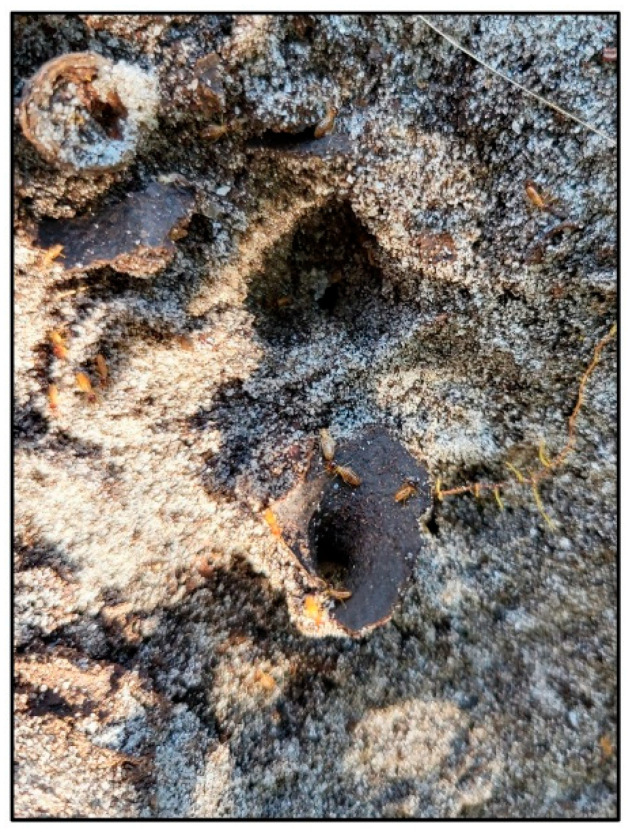
Funnel-shaped opening of carton tunnel material on the ground surface, hidden below (now removed) litter. Just below the center of the photo termites are active moving in and out of the tunnel leading down into soil, connecting to an underground nest.

**Figure 5 insects-16-01262-f005:**
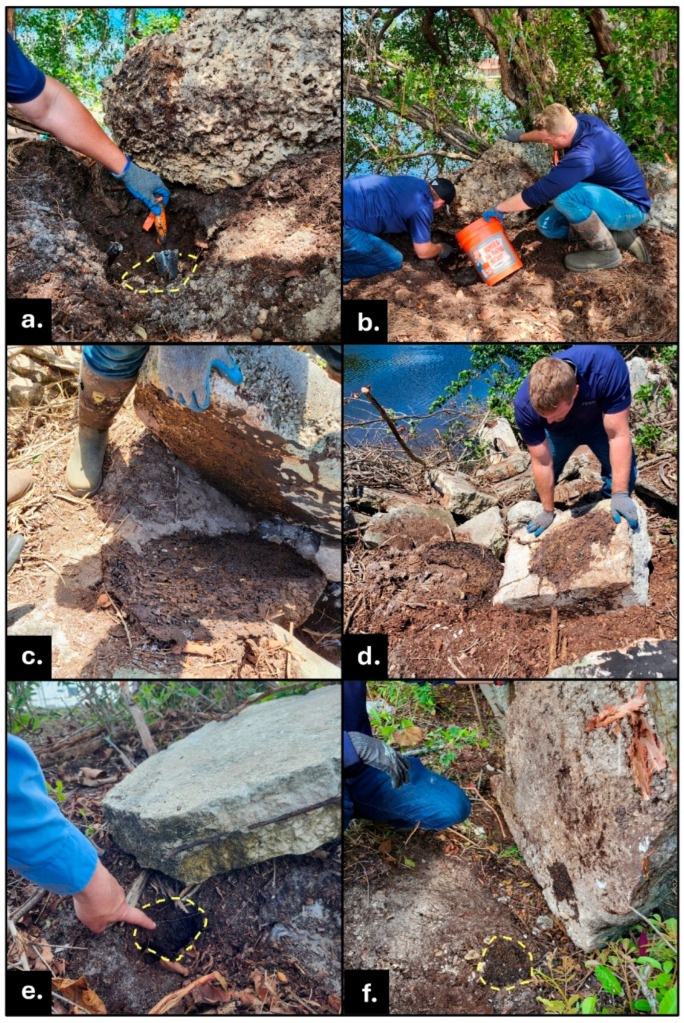
Completely underground *N. corniger* nests discovered under surfaces including big rocks and cement slabs. Nest tops abutting the underside of hard substrates are highlighted with yellow dashed lines; the remainder of each nest is underground. (**a**) Deep underground nest found after lifting large limestone boulder; same nest during removal process (**b**). (**c**) Large underground nest’s top surface exposed under concrete riprap slab. (**d**) Nest surface exposed under concrete riprap slab; canal water shown in background. (**e**) Carton discovered in small void under large block of concrete. (**f**) Small young nest found embedded in soil under large concrete slab.

**Table 1 insects-16-01262-t001:** Underground depth of *N. corniger* nests found in Broward County, Florida, in 2025. Range of depths of the base (bottom) and top of underground *N. corniger* nests discovered in Broward County, FL. Shapes and sizes of individual nests vary; this Table informs regarding the range of underground depth positions. For example, one small nest was built with its top at soil surface level and remaining carton constructed 2.5 cm below ground, rendering a shallow disk-shape. The deepest discovered ‘top’ of a nest was 45.7 cm below the soil surface.

Underground Nest Type	Number of Nests	Depth Range (Minimum to Maximum, cm) of Nest Base Relative to Soil Surface	Depth Range (Minimum to Maximum, cm) of Nest Top Relative to Soil Surface	Number of Nests Found with Apex Under and Abutting Non-Wood Substrates (e.g., Concrete, Rock)
Completely Underground	20	2.5 cm to 55.9 cm below soil surface	2.5 cm above to 45.7 cm below soil surface	8
Partially Underground	13	7.6 cm to 30.5 cm below soil surface	15.2 cm above to 5.1 cm below soil surface	2

## Data Availability

The original contributions presented in this study are included in the article/Appendix A. Further inquiries can be directed to the corresponding author.

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
