# Peer review of "Underground Nests and Foraging Activity of Invasive Conehead Termites (*Nasutitermes corniger*; Blattodea: Termitidae)"

_insects, 2025, doi:10.3390/insects16121262_

Round 1
Reviewer 1 Report
Comments and Suggestions for Authors
The manuscript "Underground nests and foraging activity of invasive conehead termites (Nasutitermes corniger; Blattodea: Termitidae) reports astonishing information and circumstances pertaining to an invasive insect that exhibits behavior contrary to established information. The manuscript should garner considerable attention from entomologists.
Authors should number the nests detailed in their results using a simple-to-follow designation like a number or number/ letter combination. It is hard to follow the narrative on sites and nests.
Consistency with terms should be checked throughout the paper.
Lines 286, 330, 340 and other locations.
There are terms such as “alate nymphs” that might reference different terms like “late nymph” or “mature and maturing alates”?
Another example includes the terms “Immatures” and “larvae”?
Some of the narrative could be abbreviated to provide clarity especially those sentences that start with prepositional phrases.
Lines – 336-338 “By using hands or gentle probing tools to discern differential densities of surrounding loose soil, then encountering the harder nest carton, the precise location of the structure was revealed.”
Suggest.
The precise location of the nest was revealed by using hands or gentle probing with tools to discern soil density differences attributed to galleries leading to the harder nest carton.
Lines 338-241 “The basketball-sized nest extracted from under the soil level (Figure 3a; Figure 4) was heavily embedded with twigs, roots and leaf debris. It housed at least 9 alate-derived Queens, mature and maturing alates, eggs and immatures, and a lively abundance of workers and soldiers.”
Suggest.
The basketball-sized nest, [ID whatever], embedded with twigs, roots and leaf debris was located (??_cm) below the soil surface (Figure 3a; Figure 4). That nest housed 9 alate-derived queens along with alates, late-stage nymphs, eggs, larvae, workers and soldiers.
Line 343-350 No reason to repeat information in the first 3 sentences of this section that could be listed in lines 117-165.
“The third, isolated site (location noted in section 2.1) is directly across the Dania Cut- Off Canal from the Airport site. A single, isolated nest was found on an overgrown, 12 hectare property that our eradication program identified as high risk for N. corniger colonization given its proximity to known swarming over two decades.”
Suggest.
The Site ~30 m east of Interstate 95 (or whatever is ID decided) involved a nest discovered by observing active foraging tunnels on small trees with no visible aerial nests but where tunnels tracked under a concrete slab.
Reviewer 2 Report
Comments and Suggestions for Authors
This manuscript describes the subterranean nesting strategy of the invasive neotropical termite Nasutitermes corniger in Florida, USA. This is an important topic, as understanding nesting and dispersal strategies is essential for improving control measures against invasive populations of this species.
The occurrence of subterranean nests appears to be related to soil type and environmental conditions. In my previous field observations in Venezuela, I also found subterranean structures of N. corniger in mangrove areas, all originating from tree nests. In São Paulo city, Brazil, this species has been documented as infesting trees subterraneanly (https://doi.org/10.1007/s13744-014-0269-y). This suggests that N. corniger may use subterranean structures as an adaptive strategy to extend its foraging range, avoid predation by ants and birds, or escape the high surface temperatures typical of both tropical and urban environments.
Since these findings are ecologically relevant and can support management programs, I suggest improve the manuscript with a summary table containing demographic and physical information on the studied subterranean colonies. The table should include, in particular, nest size, the number and type of reproductive individuals (primary or adultoids), and the relative proportions of soldiers, juveniles, and eggs. Such information would enhance the study's significance, improve its biological interpretation, and facilitate comparisons with other Nasutitermes species in their natural and invasive ranges.
